# Experimental demonstration of quantum advantage for one-way communication complexity surpassing best-known classical protocol

Niraj Kumar[1,2], Iordanis Kerenidis[2] & Eleni Diamanti [1]

Demonstrating a quantum advantage with currently available experimental systems is of utmost importance in quantum information science. While this remains elusive for quantum computation, the field of communication complexity offers the possibility to already explore and showcase this advantage for useful tasks. Here, we define such a task, the Sampling Matching problem, which is inspired by the Hidden Matching problem and features an exponential gap between quantum and classical protocols in the one-way communication model. Our problem allows by its conception a photonic implementation based on encoding in the phase of coherent states of light, the use of a fixed size linear optic circuit, and single-photon detection. This enables us to demonstrate in a proof-of-principle experiment an advantage in the transmitted information resource over the best known classical protocol, something impossible to reach for the original Hidden Matching problem. Our demonstration has implications in quantum verification and cryptographic settings.

[1] Sorbonne Université, CNRS, LIP6, F-75005 Paris, France. [2] IRIF, CNRS, Université Paris Diderot, 75013 Paris, France. Correspondence and requests for materials should be addressed to N.K. (email: nirajhariom@gmail.com) or to I.K. (email: jkeren@irif.fr) or to E.D. (email: eleni.diamanti@lip6.fr)

A major objective in quantum information science presently is finding communication and computational tasks for which it is possible to demonstrate in practice that using quantum instead of classical resources leads to superior performance in terms of computational power, security, or communication efficiency. In the quest for such demonstrations for computational tasks[1], significant achievements include Boson Sampling[2,3], which has been implemented for small sizes[4–7], and sparse commuting (IQP) or random quantum circuits[8–14]. Another recent proposal deals with the power of quantum interactive proofs for verifying NP-complete problems with small proofs[15,16].

Concerning communication tasks, there have been several works demonstrating security impossible to achieve by classical means, including quantum key distribution[17,18] and several other cryptographic primitives in various configurations[19–24], or involving nonlocal games that rely on the violation of Bell inequalities[25–27].

In addition to increased security, quantum technologies can also provide an advantage in terms of communication and information resources, such as the amount of information that needs to be transmitted to jointly perform a distributed task between two or more parties who each receive an input, or the total time this takes. Calculating and optimizing the use of such resources is the goal of the field of communication complexity, where protocols typically either minimize the amount of information that needs to be exchanged to solve a problem with certainty, or maximize the probability of solving the problem successfully using a restricted amount of communication. This field has a great range of applications including, for instance, the optimization of very large-scale integrated circuits or data structures. It has been shown that quantum resources lead to exponential asymptotic savings compared with classical resources for several protocols[28–33], including the Hidden Matching protocol[34] used in our work. The underlying factor that enables this advantage is that while in classical networks such tasks require a very large amount of information exchange, when quantum resources are available it is sufficient for one of the parties to generate, locally manipulate, and send specific states called quantum fingerprints. However, these are highly entangled multiqubit states of large dimension, whose generation is out of reach of experimental photonic technologies currently used in quantum communications.

A significant step in the direction of experimental quantum communication complexity was made by theoretical work proposing a mapping for encoding quantum communication protocols involving pure states of many qubits, unitary operations, and projective measurements to protocols based on coherent states of light in a superposition of optical modes, linear optics operations, and single-photon detection[35]. This model was used to propose the practical implementation of coherent state quantum fingerprints for computing the equality function in the simultaneous message passing model of communication complexity[36], leading to experiments demonstrating a quantum advantage in the transmitted information in this model[37,38]. Further work proposed a model involving multiplexed coherent state fingerprints to improve not only the information resource but also the communication resource[39]. We also remark that a communication complexity advantage in time was claimed to be experimentally shown recently for the quantum switch resource used in indefinite causal structures[40].

Here, we define a communication task and experimentally demonstrate a quantum advantage over the best-known classical protocol in the one-way communication complexity model, where only one party is allowed to send a message to a second one, who outputs a solution to the task—a model particularly suitable for applications in quantum networks. More precisely, based on the Hidden Matching problem introduced in ref. [34], we define the Sampling Matching problem, for which we show that it remains a hard problem for classical one-way communication while there is a quantum protocol that is exponentially more efficient with respect to the transmitted information than any randomized classical protocol with bounded error. We then apply the aforementioned coherent state mapping to Sampling Matching and we show that its implementation in this framework requires a single beam splitter and two detectors, contrary to the original Hidden Matching problem that would require the number of active components to increase at least logarithmically with the input size of the problem. The conception of Sampling Matching was inspired by a passive implementation of the round robin differential phase shift quantum key distribution (RR-DPS-QKD) protocol[41,42], which trades simplicity and stability of the experimental setup with the need for remote phase locking in a full-scale implementation. In our case, exploiting these concepts allows us to use a state-of-the-art photonic system involving encoding in the phase of weak coherent states, linear optics, and single-photon detection, for a proof-of-principle implementation of Sampling Matching, which outperforms the best-known classical protocol with respect to the transmitted information from threshold input size of around 3000. The simplicity of our experimental demonstration paves the way to the demonstration of a number of useful communication tasks that rely on similar principles.

## Results

**Sampling matching.** We start by defining a one-way communication task that we call Sampling Matching (SM), which is inspired by the Hidden Matching (HM) problem defined in refs. [31,34] (see the Methods section for description and analysis of Hidden Matching). We then outline the linear optic circuit necessary for implementing Sampling Matching, which showcases the fact that through this problem we are able to drastically reduce the resources required for demonstrating a quantum advantage in the model of one-way communication complexity.

The Sampling Matching problem is illustrated in Fig. 1. It is a task involving two players, Alice and Bob, and is described as follows. For any positive even integer $n$, Alice receives as input a string $x \in \{0, 1\}^n$, while Bob does not receive an input. His task is to sample a perfect matching $\sigma_i$ on the complete graph of $n$ vertices (where the vertices are indexed with the numbers $\{1, 2, \ldots, n\}$ and all edges are present) uniformly at random from a set of $n-1$ edge-disjoint perfect matchings $\mathcal{M}_n \in \{\sigma_1, .., \sigma_{n-1}\}$. A perfect matching here is a list of $n/2$ pairs of vertices such that no vertex appears twice in the list. A set of edge-disjoint perfect matchings is a set of matchings where each edge (pair of vertices) appears at most once in the set. It is well known that the complete graph of $n$ nodes can be decomposed into a set of $n-1$ edge-disjoint perfect matchings. This set is known to both Alice and Bob. An example for $n = 4$ is shown in Fig. 2. The objective of the problem is for Bob to output any matching $\sigma_i \in \mathcal{M}_n$ and the parity $x_k \oplus x_l$ (where $x_k$, $x_l$ are the $k$th and $l$th bit of $x$, respectively) for some pair $(k, l)$ that belongs to the matching $\sigma_i$, with the constraint that the distribution of the matchings is uniform in $\mathcal{M}_n$ even conditioned on the message $m(x)$ sent from Alice, i.e., $\mathbb{P}(\sigma_i | m(x)) = \frac{1}{|\mathcal{M}_n|}$, $\forall \sigma_i \in \mathcal{M}_n$. This constraint of uniform matching output conditioned on Alice's message is important because otherwise Alice and Bob can trivially solve the problem by sharing a public random coin which determines the matching, and then Alice sends the parity of an edge for that specific matching to Bob. This would solve the problem with $O(1)$ transmitted information and thus becomes easy classically. Since

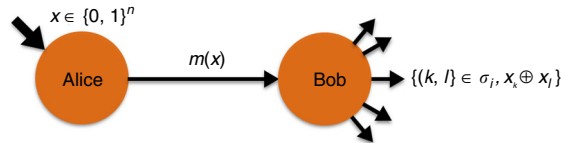

**Fig. 1** The Sampling Matching problem. Alice gets an input $x \in \{0, 1\}^n$ and sends a message $m(x)$ to Bob who outputs the pair $\langle (k, l) \in \sigma_i, b = x_k \oplus x_l \rangle$ for a matching $\sigma_i$, whose distribution is uniform in $\mathcal{M}$, even conditioned on $m(x)$. The parity should be correct with high probability for all choices of the matching

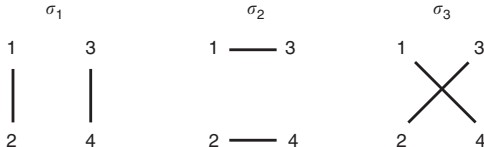

**Fig. 2** Illustration of a set of edge-disjoint perfect matchings for size $n = 4$. The matching set $\mathcal{M}_4$ has three edge-disjoint perfect matchings: $[\sigma_1:\{(1, 2), (3, 4)\}; \sigma_2:\{(1, 3), (2, 4)\}; \sigma_3:\{(1, 4), (2, 3)\}]$

we are in a communication complexity model, we expect Alice and Bob to be honest and perform the task according to the protocol.

It is also important to note that in our one-way model the communication cost of a protocol is the number of bits Alice has to send to Bob in order to solve the problem, while the transmitted information, instead of the number of bits sent, calculates the real bits of information about the inputs that the messages carry. For example, if Alice always sends the same, long message, independent of her input, then the communication cost will be large, while the transmitted information will be zero, since no information about her input has been transmitted. Transmitted information is a resource that is important for privacy, when on top of having an efficient protocol, we want the players to solve the task without learning much about the other player's input. One can define the transmitted information as the mutual information between the messages and the inputs and can upper bound it with the logarithm of the number of different messages. The transmitted information is always at most the communication cost, since one bit carries at most one bit of information, and hence the bottleneck is always the time. Last, we remark that we define our problem in the randomized setting where Bob is allowed to use random coins and output the correct value with high probability.

Let us now analyse the Sampling Matching problem in detail and show that there is an exponential gap between the classical and quantum transmitted information resource, as it is the case for the Hidden Matching problem as well. The two problems are in fact classically equivalent in their complexity. Indeed, it is relatively straightforward to see that Sampling Matching, which is effectively a sampling problem where Bob uniformly samples a matching from a set $\mathcal{M}_n$ and then uses Alice's message to find the parity of an edge in the matching, and Hidden Matching, where Bob a priori receives a uniformly random matching from the set as input, are effectively equivalent with respect to their complexity.

- SM → HM: Imagine there exists a protocol for Sampling Matching, meaning Alice sends a message $m$ and Bob samples uniformly a matching $\sigma_i$ from all matchings and uses $m$ to output a parity of an edge in $\sigma_i$. Then, Bob uses the same protocol until the output of his sampling is the matching that

he has received as input, in which case he computes the parity and outputs as in the Sampling Matching protocol. The error in HM is the same as in SM.

- HM → SM: Imagine there exists a protocol for Hidden Matching. Then, to solve Sampling Matching, Bob first samples a matching uniformly at random, and then Alice and Bob use the protocol for HM and output accordingly. The error is the same.

The classical complexity bound for Sampling Matching is $\Omega(\sqrt{n})$ and is the same as the one for the Hidden Matching problem (see the Methods section for details). More specifically, in order for Bob to succeed with an error probability $p_{\text{error}}$, we must have for the size of Alice's message that

$$c \geq \frac{\log_2 e}{e}\left(\frac{1}{2} - p_{\text{error}}\right)\sqrt{n-1}. \tag{1}$$

This bound was proven to be tight by describing a randomized one-way protocol using the birthday paradox argument to show that only $\mathcal{O}(\sqrt{n})$ classical bits are sufficient to solve the problem. In particular, for $p_{\text{error}} \leq 0.1$, the communication message size for the best-known classical protocol is $c \geq \sqrt{2\log_e 10}\sqrt{n}$. This bound as well as the lower bound of Eq. (1) will be used later in the performance analysis of our scheme.

When quantum resources are available, the task can be solved by transmitting an exponentially smaller number of qubits, similarly to Hidden Matching (see the Methods section for details). Alice encodes her $n$-bit input $x$ into the state

$$|x\rangle = \frac{1}{\sqrt{n}}\sum_{k=1}^{n}(-1)^{x_k}|k\rangle, \tag{2}$$

where $x_k$ is the $k$th bit of the string $x$, and sends it to Bob. This state $|x\rangle$ is referred to as the fingerprint of the input $x$. Bob uniformly picks a matching $\sigma_i \in \mathcal{M}_n$ and then measures the state $|x\rangle$ in the basis $\left\{\frac{1}{\sqrt{2}}(|k\rangle \pm |l\rangle)\right\}$, with $(k, l) \in \sigma_i$ to output the pair $\langle (k, l), b = x_k \oplus x_l \rangle$ with certainty. This is because the measurement outcome $\frac{1}{\sqrt{2}}(|k\rangle + |l\rangle)$ occurs if and only if $x_k \oplus x_l = 0$, whereas $\frac{1}{\sqrt{2}}(|k\rangle - |l\rangle)$ occurs if and only if $x_i \oplus x_j = 1$. This protocol uses only $\log_2 n$ qubits, and hence both the communication and the transmitted information are exponentially better than in the classical case, where both resources must be at least $\Omega(\sqrt{n})$.

The physical implementation of the qubit protocol is extremely challenging due to the high dimensionality of the fingerprint states required to show a quantum advantage, which means that highly entangled states of many qubits need to be generated and maintained during the entire run of the protocol. Applying the coherent state mapping proposed by Arrazola and Lütkenhaus[35], it is possible to describe an alternative quantum protocol based on coherent state fingerprints[36] as follows. Alice prepares the message $|\alpha_x\rangle$, by applying the displacement operator $\hat{D}_x(\alpha) = \exp(\alpha \hat{a}_x^\dagger - \alpha^* \hat{a}_x)$ to the vacuum state, where $\hat{a}_x = \frac{1}{\sqrt{n}}\sum_{k=1}^{n}(-1)^{x_k + \phi}\hat{a}_k$ is the annihilation operator of the entire coherent state mode, and $\hat{a}_k$ is the photon annihilation operator of the $k$th time mode; she also adds an additional constant factor of $\phi \in \{0, 1\}$ chosen uniformly randomly. Hence,

$$|\alpha_x\rangle = \hat{D}_x(\alpha)|0\rangle = \bigotimes_{i=1}^{n}\left|(-1)^{x_i \oplus \phi}\frac{\alpha}{\sqrt{n}}\right\rangle_i, \tag{3}$$

where $\left|(-1)^{x_k \oplus \phi}\frac{\alpha}{\sqrt{n}}\right\rangle_k$ is a coherent state with amplitude $\frac{\alpha}{\sqrt{n}}$ occupying the $k$th time mode. Here, $|\alpha_x\rangle$ is the fingerprint for input $x$, and can be thought of as a sequence of $n$ coherent pulses

with the total mean photon number over the sequence being $\mu = \sum_k |\frac{\alpha}{\sqrt{n}}|^2 = |\alpha|^2$, which is independent of the input size.

Note that this protocol takes time $n$, since we have a sequence of $n$ time modes, and thus loses any advantage compared with the classical protocol in terms of communication time. Nevertheless, the information transmitted by this protocol remains only logarithmic, which is exponentially better that the classical protocol that requires $O(\sqrt{n})$ bits of information.

Let us now see how Alice and Bob could implement this protocol in practice. An illustration for any $n$ is shown in Fig. 3. Alice sends the state $|\alpha_x\rangle$ as described above, and Bob generates locally a sequence of $n$ coherent pulses $|\beta\rangle = \otimes_{i=1}^{n} |\frac{\alpha}{\sqrt{n}}\rangle_i$, interferes them sequentially in a balanced beam splitter with the corresponding pulses from Alice, and observes the clicks on the single-photon detectors that we name $D_0$ and $D_1$.

In the ideal setting, the state in the incoming modes of the beam splitter at the $k$th time slot is,

$$\left|(-1)^{x_k \oplus \phi}\frac{\alpha}{\sqrt{n}}\right\rangle_i \otimes \left|\frac{\alpha}{\sqrt{n}}\right\rangle_k, \tag{4}$$

and the output state is,

$$\left|\frac{(1+(-1)^{x_k \oplus \phi})}{\sqrt{2}}\frac{\alpha}{\sqrt{n}}\right\rangle_{D_0} \otimes \left|\frac{(1-(-1)^{x_k \oplus \phi})}{\sqrt{2}}\frac{\alpha}{\sqrt{n}}\right\rangle_{D_1}. \tag{5}$$

From this equation, we see that $D_0$ clicks only if $x_k \oplus \phi = 0$, while $D_1$ clicks only if $x_k \oplus \phi = 1$. Now suppose Bob gets the clicks at $k$th and $l$th time slots in detectors $D_0$ and $D_1$, respectively. This implies $x_k \oplus \phi = 0$, while $x_l \oplus \phi = 1$. Combining them results in $x_k \oplus x_l = 1$ since $2\phi \equiv 0 \pmod 2$. Therefore, Bob successfully outputs the pair $\langle (k, l) \in \sigma_i, b = x_k \oplus x_l \rangle$ for the matching $(k, l) \in \sigma_i$. This protocol only lets Bob obtain the parity information of the bits and not the bit values $x_k, x_l$ because of the hiding factor $\phi$.

The cases where Bob can make an error in inferring the correct parity value of any matching are as follows. (i) Bob does not observe any single click over the entire run of the experiment. The probability of this happening is $p_{\neg 1} = \exp(-2|\alpha|^2)$. Bob's error probability in this case is $\frac{1}{2}p_{\neg 1}$. (ii) Bob observes exactly one single click over the entire run of the experiment. Since the parity of a pair is inferred from the clicks at two distinct time slots, in this case Bob does not infer any parity outcome with certainty. The probability of exactly one single click happening is,

$$p_1 = \binom{n}{1} p_c (1 - p_c)^{n-1}, \tag{6}$$

where $p_c = 1 - \exp\left(-2\frac{|\alpha|^2}{n}\right)$ is the probability of getting a click in one time slot. Bob's error probability in this event would be $\frac{1}{2}p_1$. Combining the two cases, Bob's error probability is,

$$p_{\text{error}} = \frac{1}{2}(p_0 + p_1). \tag{7}$$

In a practical setting, we need to take into account three main sources of error: (i) the transmission and detection loss characterized by the efficiency parameters $\eta_{\text{channel}}$ and $\eta_{\text{det}}$, respectively; modeling the detection loss with a beam splitter followed by perfect detection allows us to lump these two loss factors into a single parameter $0 \le \eta \le 1$; (ii) the limited interference visibility $0 \le \nu \le 1$; and (iii) the detector dark counts characterized by the probability $p_{\text{dark}}$. As we will justify in the following, in our experimental conditions the dark count probability is negligible compared to the expected signal count probability, therefore we do not consider the effect of dark counts in our analysis. Considering experimental imperfections ($\eta$, $\nu$),

the incoming state at the $k$th time slot becomes,

$$\left|(-1)^{x_k \oplus \phi}\sqrt{\frac{\eta}{n}}\alpha\right\rangle_k \otimes \left|\sqrt{\frac{\eta}{n}}\alpha\right\rangle_k, \tag{8}$$

and the output state is now written as,

$$\left|\left(\frac{(1+(-1)^{x_k \oplus \phi})}{\sqrt{2}}\sqrt{\nu} + \frac{(1-(-1)^{x_k \oplus \phi})}{\sqrt{2}}\sqrt{1-\nu}\right)\sqrt{\frac{\eta}{n}}\alpha\right\rangle_{D_0,k} \otimes$$
$$\left|\left(\frac{(1-(-1)^{x_k \oplus \phi})}{\sqrt{2}}\sqrt{\nu} + \frac{(1+(-1)^{x_k \oplus \phi})}{\sqrt{2}}\sqrt{1-\nu}\right)\sqrt{\frac{\eta}{n}}\alpha\right\rangle_{D_1,k}. \tag{9}$$

We see that due to the limited visibility there is a nonzero click probability for the wrong detector in a given time slot. From Eq. (9), we find that the probability of a click in the correct detector at each time slot is,

$$p_c = 1 - \exp\left(-2\eta\nu\frac{|\alpha|^2}{n}\right), \tag{10}$$

while the probability that a click occurs in the wrong detector is,

$$p_w = 1 - \exp\left(-2\eta(1-\nu)\frac{|\alpha|^2}{n}\right). \tag{11}$$

Now we look at the cases where Bob can output the incorrect parity outcome: (i) He does not observe at least two single clicks in the time slots during the experiment. The probability $\mathbb{P}(\text{less than two single clicks}) = \mathbb{P}(\text{no single clicks}) + \mathbb{P}(\text{exactly one single click})$,

$$p_{\neg 11} = (1 - p_1)^n + \binom{n}{1} p_1 (1 - p_1)^{n-1}, \tag{12}$$

where $p_1 = p_c(1 - p_w) + p_w(1 - p_c)$ is the probability of observing a single click in one time slot. Bob's error probability in this case is $\frac{1}{2}p_{\neg 11}$. (ii) Bob observes at least two single clicks in the time slots. He then randomly chooses any two of those single-click slots $(k, l)$ to output the parity for pair $(k, l) \in \sigma_i$. The probability that he outputs the wrong parity value is,

$$p_{11w} = \frac{2p_c(1 - p_w)p_w(1 - p_c)}{[p_c(1 - p_w) + p_w(1 - p_c)]^2}. \tag{13}$$

Combining these two cases, Bob's total error probability is,

$$p_{\text{error}} = \frac{1}{2}p_{\neg 11} + (1 - p_{\neg 11})p_{11w}. \tag{14}$$

The quantum protocol with coherent state fingerprints for Sampling Matching that we introduced above has a complexity of $\mathcal{O}(|\alpha|^2 \log_2 n)$ for the transmitted information, where $\mu = |\alpha|^2$ is the total mean photon number in the coherent fingerprint of Alice and is independent of $n$. Note that our protocol offers an exponential advantage for the information resource, but not for the communication resource which is $n$. This is the same as in previous works on protocols with coherent states[35].

In order to illustrate the performance of this protocol for Sampling Matching with respect to the classical bounds and examine the possibility of demonstrating a quantum advantage in practice, we compare the transmitted information resource for the quantum protocols with coherent states for a given error probability $p_{\text{error}}$. The results are shown in Fig. 4a for $p_{\text{error}} = 0.1$ for the best-known classical protocol and for the coherent state protocol in the ideal and practical settings, where in the latter case we have considered the experimental parameters of Table 1. In both cases, we have found the optimal $|\alpha|^2$ for our fixed $p_{\text{error}}$ value. We have also included in the graph the classical lower bound described previously and we have additionally considered the case where Bob only outputs the parity outcome when he he obtains at least two single clicks in the experimental run, which we call the post-selected protocol.

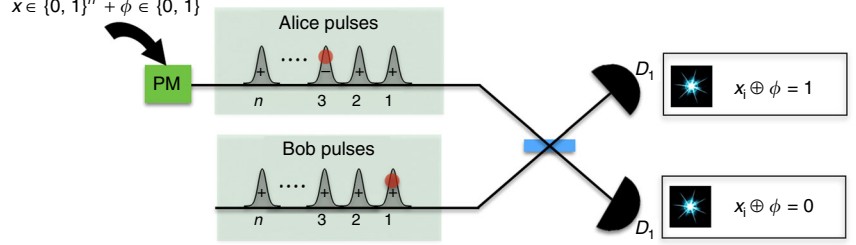

**Fig. 3** Circuit illustration for the implementation of Sampling Matching using coherent states for any input size $n$. Alice prepares her message by encoding her input $x \in \{0, 1\}^n$ and an additional factor $\phi \in \{0, 1\}$ on the phase of a train of coherent states, using a phase modulator (PM), to produce the coherent state fingerprint $|\alpha_x\rangle$. Bob, on his side, produces a sequence of $n$ states with the same total mean number as Alice's state, interferes his pulses with Alice's sequentially on a balanced beam splitter, and obtains the parity information from the clicks on the single-photon detectors $D_0$ and $D_1$. As an example, the red dots in the first and third time slots of Alice's and Bob's sequences, respectively, indicate that Bob observes a single click at $D_1$ and $D_0$ detectors respectively for these time slots, and thus he outputs $x_1 \oplus x_3 = 1$. If he obtains single clicks at more than two time slots, then he randomly chooses any two of them to output the parity outcome

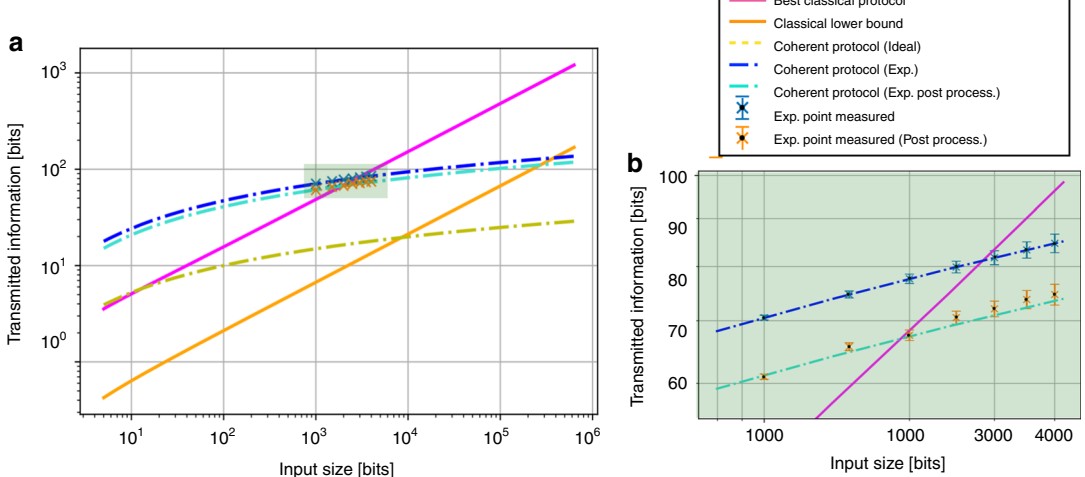

**Fig. 4** Log–log plot of transmitted information resource vs. the input size $n$ for solving the Sampling Matching problem within error probability $p_{error} = 0.1$. In panel **a**, we compare the optimal classical protocol, the classical lower bound, and the quantum protocols in the ideal setting, in the practical setup with the experimental parameters of Table 1, and in the post-selected case where Bob only outputs a parity outcome when he obtains at least two single clicks in the protocol run. For the last two protocols, we also show the experimental results obtained with the setup of Fig. 5, for input sizes between 1000 and 4000. These results are also shown more clearly in panel **b** that focuses on this region. The optimal mean photon number per pulse in each case, as well as other parameters, is given in Table 2. The error bars for the experimental points refer to the standard deviation, which primarily comes from the error in estimating the mean photon number per pulse. We see that for the experiments implementing the standard and post-selected protocols, for input size above 3000 and 2000, respectively, our results outperform the best classical protocol hence demonstrating the obtained quantum advantage

| Table 1 Experimental parameters corresponding to our implementation and used in the simulations | | | |
| --- | --- | --- | --- |
| $\eta_{channel}$ | $\eta_{det}$ | $\nu$ | $p_{dark}$ |
| 45% | 25% | $(98.8 \pm 0.3)\%$ | $(2.3 \pm 0.2) \times 10^{-6}$ |

We remark that although the ideal protocol can outperform the best classical protocol for relatively low input size, in the realistic case this can happen for $n \sim 3000$. We also observe that the post-selected case diminishes slightly this threshold and that beating the classical lower bound requires a very large input size. These threshold input sizes are similar to Hidden Matching (see the Methods section for details). However, although reaching such values would be a formidable challenge for Hidden Matching, Sampling Matching allows by its conception to reach the

threshold for the best classical protocol in practice, as we will see below.

**Experimental implementation.** The experimental setup realizing in practice the schematic illustration of Fig. 3 and that we use for our proof-of-principle implementation of the Sampling Matching problem is shown in Fig. 5.

The coherent light is generated using a low line-width (~10 kHz) continuous wave laser source operating at telecommunication wavelength (Laser1, Pure Photonics, $\lambda = 1563$ nm). An amplitude modulator (AM) is then used to produce a sequence of coherent pulses with a 1-MHz repetition rate and pulse duration of 16 ns. A balanced 50:50 beam splitter (BS1) is used to monitor the power of the laser pulse, and we use a variable optical attenuator (VOA) to attenuate the pulses to the desired mean photon number. A second 50:50 beam splitter (BS2) splits the coherent pulses in two paths, sent to Alice and Bob. We

**Fig. 5** Experimental setup for implementation of the quantum protocol for Sampling Matching with coherent states. A continuous wave laser operating at $\lambda = 1563$ nm (Laser1) followed by an amplitude modulator (AM) and an optical variable attenuator (VOA) is used for the generation of coherent light pulses at 1 MHz repetition rate and with 16 ns duration, at the mean photon number required for the protocol (see main text for details). The pulses are split at beam splitter BS2 to two paths corresponding to Alice and Bob. Alice encodes the phase information to her pulses sequentially according to her input string $x \in \{0, 1\}^n$ using a phase modulator (PM), while Bob prepares his sequence by encoding 0 to his pulses. Both modulators are controlled by a data acquisition card (DAQ). We use a delay line (DL) to adjust precisely the path lengths of the sequences such as to optimize their interference at the balanced beam splitter BS3. The output pulses are then directed to two single-photon detectors $D_0$ and $D_1$, and the detection events are registered using a time tagger. To monitor and correct the phase drift in the pulse sequences of Alice and Bob, we use a phase-correction loop, which consists of a second continuous wave laser operating at $\lambda = 1527$ nm (Laser2), followed by amplitude modulation and attenuation, and a combination of circulators (C1, C2), an optical filter (OF) and a photodiode (PD), to suitably direct the monitoring pulses through the setup in the opposite direction than the signal while preventing this light from reaching unwanted devices. We also compensate for Alice's and Bob's path length difference induced by the presence of a different number of components

introduce a delay line (DL, Kylia) to fine tune the path lengths of Alice and Bob, hence ensuring that their pulses arrive simultaneously at the 50:50 beam splitter BS3 and interfere optimally. Before this, Alice and Bob modulate their pulses, each using a phase modulator (PM). These are driven by a data acquisition card that provides the desired voltage levels, which is fixed for Bob and corresponding to her input $x \in \{0, 1\}^n$ for Alice. After interfering, the pulses are detected by telecom wavelength, free running InGaAs single-photon detectors $D_0$ and $D_1$ (ID230, IDQuantique). The detection events are recorded and analyzed with a precision of 1 ps using a time tagger (quTAG, QuTools).

The remaining components in the experimental setup are used for the phase-correction loop that we use to monitor and correct the phase drift between Alice's and Bob's pulses. More specifically, we introduce a second continuous wave laser source (Laser2, Pure Photonics, $\lambda = 1527$ nm) that is modulated similarly as described before. The pulses are directed through a circulator (C2) to BS3, where they are separated and then interfere on BS2 before being detected using a photodiode (PD). A second circulator (C1) prevents any of this light to go into the direction of Laser1. Furthermore, an optical filter (OF) is used in the path leading to detector $D_0$ to ensure that only light from Laser1 ($\lambda = 1563$ nm) reaches the detector. The difference in the length of the paths leading to $D_0$ and $D_1$ due to the presence of these components is appropriately compensated using a fiber before detector $D_1$. To correct the phase drift, we use an averaging technique that estimates the phase drift over a block of pulses and corrects accordingly the phase in the next block (see the Methods section for details).

We are now ready to analyze the performance of our experiment for Sampling Matching. The relevant experimental parameters that have also been used for the simulations are shown in Table 1. The channel transmission loss, i.e., the loss from when Alice and Bob apply their phase modulation to the input of detectors $D_0$ and $D_1$ is 3.5 dB, hence $\eta_{\text{channel}} \approx 45\%$. Furthermore, our single-photon detectors feature a quantum efficiency $\eta_{\text{det}} \approx 25\%$. The effect of these losses is that it is necessary to increase the mean photon number in the coherent

fingerprint state compared with the ideal setting, in order to achieve the desired error rate $p_{\text{error}}$.

The limited visibility, $v$, is due to the imperfect interference of Alice and Bob's pulses. It is important to remark that in our proof-of-principle implementation, we use a single laser for generating the pulses that Alice and Bob need to prepare their states, which is important for improving the visibility. However, the preparation of these states and all subsequent steps are done independently following the protocol, hence enabling us to use this setup for assessing the quantum advantage. As noted before, to achieve a high visibility, we also fine tune the delay line in the setup by following a simple calibration procedure whereby we send first sequences of 0 inputs to both Alice and Bob and then sequences of 0 and 1 inputs to Alice and Bob, respectively, and observe the resulting detector clicks.

Finally, we further investigate the dark counts to make sure it is safe to neglect them in our analysis and indeed we observe that the signal click probability is substantially (three orders of magnitude) larger than the dark count probability. We also note that our detectors feature a dead time of 10 μs, which means that after a detection event, the detector becomes idle for the next 10 pulses. For the input size targeted in our work ($\geq 1000$), the probability of a click within these pulses is extremely low due to the extremely low photon number of pulse that we use. This effect can therefore be safely neglected.

Based on these experimental parameters obtained in our setup, we estimate the optimal mean photon number $\mu$ for the entire coherent fingerprint state that achieves the desired error rate $p_{\text{error}} = 0.1$, and hence the mean photon number per pulse, $\mu_p$, for input size $n$ around the threshold regions observed in Fig. 4, in particular from 1000 to 4000. These values are summarized in Table 2. In Fig. 4a, b, we show the experimentally obtained results for the transmitted information based on the above analysis. We see that for input size above 3000, our experiment for Sampling Matching provides an advantage in information compared to the best classical protocol, even within the error bars.

Furthermore, we also consider the case when Bob runs the Sampling Matching protocol multiple times (#Runs in Table 2) and gives an output only for those runs where he gets a parity

**Table 2 Experimental parameters and analysis**

| $n$ | 1000 | 1500 | 2000 | 2500 | 3000 | 3500 | 4000 |
|---|---|---|---|---|---|---|---|
| $p_{error}$ | 0.1 | 0.1 | 0.1 | 0.1 | 0.1 | 0.1 | 0.1 |
| $\mu_p$ (*$10^{-3}$) | 7.08 ± 0.01 | 4.72 ± 0.01 | 3.54 ± 0.01 | 2.83 ± 0.01 | 2.36 ± 0.01 | 2.02 ± 0.01 | 1.77 ± 0.01 |
| #Runs | 848 | 568 | 475 | 381 | 317 | 272 | 238 |
| #Runs$_{no\ click}$ | 115 | 68 | 62 | 45 | 38 | 31 | 28 |
| #Runs$_{click}^{wrong}$ | 26 | 26 | 20 | 17 | 16 | 7 | 11 |
| $p_{error}^{POST}$ | 0.03 | 0.04 | 0.04 | 0.04 | 0.05 | 0.03 | 0.05 |
| $\mu_p^{POST}$ (×$10^{-3}$) | 6.12 ± 0.01 | 4.15 ± 0.01 | 3.08 ± 0.01 | 2.50 ± 0.01 | 2.08 ± 0.01 | 1.79 ± 0.01 | 1.56 ± 0.01 |

We perform the Sampling Matching protocol for seven different input sizes, $n$, from 1000 to 4000. The objective is to output the matching parity outcome with an error probability of at most $p_{error} = 0.1$. We run the protocol #Runs times for each input size. Out of these runs, #Runs$_{noclicks}$ is the number of cases where we do not obtain at least two single clicks. Based on this, we compute the average photon number per pulse, $\mu_p^{POST}$, in the scheme where Bob only outputs the parity outcome for those runs where he gets at least two single clicks. Finally, #Runs$_{click}^{wrong}$ is the number of cases where Bob obtains at least two single clicks and he outputs the wrong parity outcome. This determines the error rate, $p_{error}^{POST}$, after post selection

outcome. Without this post selection, every time Bob would not obtain the parity outcome, he would output a random parity with error rate 1/2. However, with post selection, since he rejects those no-parity outcome cases, he can succeed with a lower error rate $p_{error}^{POST}$. This can also be interpreted as performing the protocol with lower mean photon number,

$$\mu_p^{POST} = \mu_p \frac{(\#\text{Runs} - \#\text{Runs}_{noclicks})}{\#\text{Runs}}. \quad (15)$$

The corresponding experimental values are provided in Table 2. In Fig. 4, we also plot the experimental results for the transmitted information in the post-selected scenario. We observe that the quantum protocol performs the Sampling Matching task with lower resources than the best classical protocol from input size of 2000 and above, hence demonstrating a quantum advantage in this case as well.

## Discussion

The results that we have presented demonstrate rigorously a quantum advantage in the information resource in the one-way model of communication complexity. We achieved this by introducing the Sampling Matching problem, which is inspired by the emblematic Hidden Matching problem, and by analyzing it using the recently formulated coherent state mapping for quantum communication protocols. These two advancements enabled us to bypass the great challenge associated to the implementation of such tasks with the usual high dimensional multi-qubit fingerprint states.

As we have noted, an essential element of our proof-of-principle implementation is the ability to achieve high interference visibility, which has been facilitated in our case by the use of a single laser for generating the coherent states used by Alice and Bob for their sequences and the fine tuning of the path length difference using a delay line. In a full-scale implementation, where two separate lasers would be used, maintaining a good interference would require the use of stable, ultra narrow linewidth lasers such that the phase difference between the pulses would be slower than the duration of the experimental run[43]. In combination with phase-correction techniques like the one used in our experiment, such an experiment is foreseeable in the near future and would be useful more generally for quantum communication tasks.

We also remark that our experimental results allow outperforming the best-known classical protocol but not the classical lower bound. For this, we need an input size on the order of $\sim 10^6$, which in turn would require attenuating the coherent pulses to a mean photon number per pulse of the same order. In this case, the dark counts of the single-photon detectors cannot be neglected any longer; indeed, the dark

count rate exhibited by the detectors used in our experiment is precisely of this order, and therefore it is impossible to show an advantage due to the noise. However, this would become possible using ultra low dark count superconducting nanowire single-photon detectors[44], which also feature good quantum efficiencies and are commercially available.

The Sampling Matching problem that we have defined can also be seen as a verification tool, with applications in cryptographic and computational settings, most notably quantum money schemes, where it can replace the verification techniques that use Hidden Matching[45,46]. The soundness of verification in these schemes depends on the size of the input, hence since Sampling Matching allows for a simple implementation for large input sizes, our approach may readily increase the robustness of these schemes.

## Methods

**Hidden matching.** The intuition behind introducing the Sampling Matching problem was the Hidden Matching problem. Here, we describe this problem as defined in refs [31,34] and provide a possible linear optic implementation, which as we will see is out of reach for current experimental technology.

The Hidden Matching problem is illustrated in Fig. 6. Here, for any positive even integer $n$, Alice receives as input a string $x \in \{0, 1\}^n$ while Bob receives a perfect matching $\sigma_i$ on the complete graph of $n$ vertices uniformly at random from a set of $n - 1$ edge-disjoint perfect matchings $\mathcal{M}_n \in \{\sigma_1, .., \sigma_{n-1}\}$. The objective of the problem is for Bob to output any one of the $n/2$ possible parity values $x_k \oplus x_l$ for some pair $(k, l)$ that belongs to the matching $\sigma_i$ with minimum communication and information cost. It is important to note that we look at the one-way communication model for this problem, otherwise it is easy to see that the task can be done with logarithmic communication, since Bob can send to Alice the indices $(k, l)$ and Alice will reply with the parity. Furthermore, as for Sampling Matching, we analyse this problem in the randomized setting where Bob is allowed to use random coins and output the correct value with high probability.

Bar-Yossef et al.[34] and later Buhrman et al.[47] showed that the best classical protocol for Hidden Matching must have communication and transmitted information of at least $\Omega(\sqrt{n})$. The detailed proofs can be found in refs [34,47] but we provide here a high-level description. The main idea is that Alice's message should allow Bob to output the parity of an edge from each one of the possible matchings, in other words for $O(n)$ different edges, since there are $O(n)$ edge-disjoint matchings in the set. No matter which $\Omega(\sqrt{n})$ edges one picks on a complete graph of $n$ nodes, they will always contain at least $\Omega(\sqrt{n})$ different vertices corresponding to different bits of the input $x$, and hence Alice must send at least $\Omega(\sqrt{n})$ bits of information about these bits if Bob is to be able to solve the problem and hence also $\Omega(\sqrt{n})$ communication (since communication is always at least as much as the information). The proof structure for computing the lower bound is as follows[47]: if Alice's message to Bob is small, let's say $c$ bits, then the set of inputs $x \in \{0, 1\}^n$ for which Alice sends a particular message $m$ will be large (typically of the order of $2^{n-c}$). This would mean that Bob will have very little knowledge for most of the bits of $x$. Using techniques from ref. [48] this implies that

Bob would not be able to correctly answer the parity $x_k \oplus x_l$ for most of the $\binom{n}{2}$

possible pairs $(k, l)$. Even though Bob has some relaxation in the sense that he can output the parity outcome of any one of the $n/2$ pairs of $\sigma_i$, still it turns out that on average it is hard for him to output the correct parity outcome. Using this idea, it was shown in[47] that in order for Bob to succeed with an error probability $p_{error}$, we

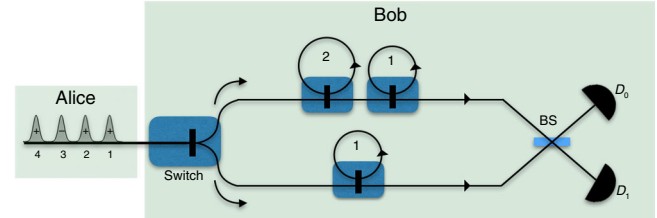

**Fig. 6** The Hidden Matching problem. Alice receives an input $x \in \{0, 1\}^n$ while Bob receives as input a matching $\sigma_i$ uniformly at random from an edge-disjoint set $\mathcal{M}_n \in \{\sigma_1, ..\sigma_{n-1}\}$. The objective of the problem is for Bob to output the correct parity value, $b$, for any one of the pairs in the matching $\langle (k, l) \in \sigma_i, b = x_k \oplus x_l \rangle$. Only one-way communication, from Alice to Bob is allowed, in the form of a message $m(x)$

must have for the size of Alice's message that

$$c \geq \frac{\log_2 e}{e}\left(\frac{1}{2} - p_{\text{error}}\right)\sqrt{n-1}. \tag{16}$$

Bar-Yossef et al. also proved that this bound is tight by describing a randomized one-way protocol using the birthday paradox argument to show that only $\mathcal{O}(\sqrt{n})$ classical bits are sufficient to solve the problem. The proof structure is as follows: Note that Bob's matching belongs to the set $\mathcal{M}_n$ of $n-1$ edge-disjoint perfect matchings. Since Alice has no information about which matching Bob has received, to maximize the probability of success she encodes her message to contain the parity information of at least one pair from each matching with high probability. Suppose she does this by sending $c$ random bits of the input $x$ or equivalently $c(c-1)/2$ pairs to Bob. Each perfect matching $\sigma_i$ that Bob would receive has $n/2$ pairs. Thus the matching set $\mathcal{M}_n$ has in total $n(n-1)/2$ distinct pairs, since each edge appears only once. The probability that none of the pairs that Alice sends to Bob is in the matching $\sigma_i$ received by Bob is,

$$p_{\text{error}} = \left(1 - \frac{1}{n-1}\right)^{c(c-1)/2} \approx \exp(-c^2/2n). \tag{17}$$

For $p_{\text{error}} \leq 0.1$, the communication message size for the best-known classical protocol is therefore $c \geq \sqrt{2\log_e 10}\sqrt{n}$. These bounds are the same for Sampling Matching as explained previously.

Using quantum resources, the above task can be solved by transmitting an exponentially smaller number of qubits[34]. Alice encodes her input $x$ into the fingerprint state, $|x\rangle = \frac{1}{\sqrt{n}}\sum_{k=1}^n (-1)^{x_k}|k\rangle$, where $x_k$ is the $k$th bit of the string $x$, and sends it to Bob. For any matching $\sigma_i \in \mathcal{M}$ that Bob has as input, there exists a measurement by Bob which allows him to give the correct answer with certainty. To do so, he just measures the quantum state in the basis $\left\{\frac{1}{\sqrt{2}}\left(|k\rangle \pm |l\rangle\right)\right\}, \forall (k, l) \in \sigma_i$. The outcome $\frac{1}{\sqrt{2}}(|k\rangle + |l\rangle)$ occurs if and only if $x_k \oplus x_l = 0$ whereas $\frac{1}{\sqrt{2}}(|k\rangle - |l\rangle)$ occurs if and only if $x_i \oplus x_j = 1$. Thus Bob gets the parity result of one of the pairs $(k, l) \in \sigma_i$ with certainty. This protocol uses only $\log_2 n$ qubits, and hence both the communication and the transmitted information are exponentially better than in the classical case.

As we did for Sampling Matching, let us now analyze the physical implementation of the Hidden Matching problem under the coherent state framework of refs. [35,36]. In this framework, Alice prepares the coherent state fingerprint as a sequence of $n$ coherent pulses whose phase corresponds to her input $x \in \{0, 1\}^n$, hence,

$$|\alpha_x\rangle = \overset{n}{\underset{k=1}{\otimes}}\left|(-1)^{x_k}\frac{\alpha}{\sqrt{n}}\right\rangle_k, \tag{18}$$

where $\mu = |\alpha|^2$ is the mean photon number for the state $|\alpha_x\rangle$, which is independent of the input size $n$. As shown in Fig. 7, which illustrates how this scheme could be implemented in practice for $n = 4$, upon receiving the state $|\alpha_x\rangle$ from Alice, Bob rearranges the input modes of $|\alpha_x\rangle$ according to the pairs $(k, l) \in \sigma_i$ using a number of switches and delay lines, interferes all the pairs in $\sigma_i$ sequentially in a balanced beam splitter, and observes the clicks recorded by single-photon detectors $D_0$ and $D_1$.

In the ideal setting, the state in the incoming modes at the beam splitter for pairs $(k, l)$ is,

$$\left|(-1)^{x_k}\frac{\alpha}{\sqrt{n}}\right\rangle_k \otimes \left|(-1)^{x_l}\frac{\alpha}{\sqrt{n}}\right\rangle_l, \tag{19}$$

and following the standard beam splitting transformations the state at the output

**Fig. 7** Circuit illustration for the implementation of Hidden Matching using coherent states, for matchings from the set in Fig. 2. Alice encodes her input $x \in \{0, 1\}^4$ as a train of four pulses and sends it Bob. Depending on his input matching $\sigma_i \in \mathcal{M}_4$, Bob uses a switch to send each of the pulses in the coherent state sequence in the upper or the lower arm. Both arms contain an appropriate combination of switches and delay lines, where the number indicated in each loop denotes the number of time steps the loop will delay the corresponding pulse and one step is equal to the duration between the pulses in the sequence. The number of active elements needed to implement the protocol is 4. For a general input size $n$, this number grows as $\mathcal{O}(\log n)$

modes is,

$$\left|\frac{1 + (-1)^{x_k \oplus x_l}}{\sqrt{2}}\frac{\alpha}{\sqrt{n}}\right\rangle_{D_0} \otimes \left|\frac{1 - (-1)^{x_k \oplus x_l}}{\sqrt{2}}\frac{\alpha}{\sqrt{n}}\right\rangle_{D_1}. \tag{20}$$

From the above equation, we see that $D_0$ clicks only if $x_k \oplus x_l = 0$, and $D_1$ clicks otherwise. Now if Bob gets clicks at multiple time slots, he picks arbitrarily one of these time slots and outputs the pair $\langle (k, l) \in \sigma_i, b = x_k \oplus x_l \rangle$ depending on which detector clicked. The only way he can output an incorrect parity value is if he does not observe any click during the entire run of the protocol, which happens with probability $p_0 = \exp(-|\alpha|^2)$, in which case he outputs a random choice. Thus his error probability is $p_{\text{error}} = \frac{1}{2}p_0$.

In a practical setting, and following the same model for experimental imperfections as for Sampling Matching, the incoming state becomes,

$$\left|(-1)^{x_k}\sqrt{\frac{\eta}{n}}\alpha\right\rangle_k \otimes \left|(-1)^{x_l}\sqrt{\frac{\eta}{n}}\alpha\right\rangle_l, \tag{21}$$

and the output state is now written as,

$$\left|\left(\frac{(1+(-1)^{x_k \oplus x_l})}{\sqrt{2}}\sqrt{\nu} + \frac{(1-(-1)^{x_k \oplus x_l})}{\sqrt{2}}\sqrt{1-\nu}\right)\sqrt{\frac{\eta}{n}}\alpha\right\rangle_{D_0} \otimes$$
$$\left|\left(\frac{(1-(-1)^{x_k \oplus x_l})}{\sqrt{2}}\sqrt{\nu} + \frac{(1+(-1)^{x_k \oplus x_l})}{\sqrt{2}}\sqrt{1-\nu}\right)\sqrt{\frac{\eta}{n}}\alpha\right\rangle_{D_1}. \tag{22}$$

From the above equation, we see that the probability that there is a click in the correct detector is,

$$p_c = 1 - \exp\left(-2\eta\nu\frac{|\alpha|^2}{n}\right), \tag{23}$$

while the probability that the wrong detector clicks is,

$$p_w = 1 - \exp\left(-2\eta(1-\nu)\frac{|\alpha|^2}{n}\right). \tag{24}$$

Let us now consider the cases where Bob can output an incorrect parity value outcome. (i) He does not observe any single click over the entire run of the experiment. The probability of this happening is $p_{\neg 1} = (1 - p_1)^{n/2}$, where $p_1 = p_c(1-p_w) + p_w(1-p_c)$ is the probability of observing a single click in one time slot. In this case, he outputs a random parity value. (ii) Bob observes at least one single click within all time slots. He then randomly chooses any one of those to output the parity value. The probability that he outputs the wrong parity value is $p_{1w} = \frac{p_w(1-p_c)}{p_w(1-p_c) + p_c(1-p_w)}$. From these two cases, we find that Bob's error probability is,

$$p_{\text{error}} = \frac{1}{2}p_{\neg 1} + (1 - p_{\neg 1})p_{1w}. \tag{25}$$

The quantum protocol with coherent state fingerprints for Hidden Matching that we have described and analyzed has a complexity of $\mathcal{O}(|\alpha|^2 \log_2 n)$ for the transmitted information, where $\mu = |\alpha|^2$ is the total mean photon number in the coherent state. Note again that the exponential advantage concerns the information, but not the communication resource. As for Sampling Matching, for a fixed error probability $p_{\text{error}} = 0.1$, we calculate the optimal $|\alpha|^2$ and the transmitted information for the quantum protocols with coherent states for the ideal, practical and post-selected cases. The latter here refers to the case where Bob only outputs the parity outcome when he observes at least one click in the protocol run. The results are shown in Fig. 8, where we have also included the bounds for the best classical protocol and the classical lower bound. The same remarks on the

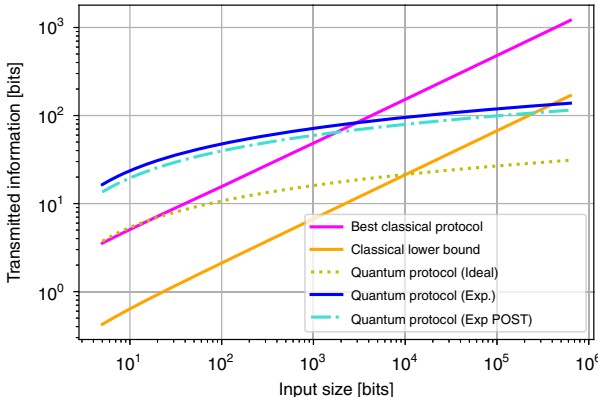

**Fig. 8** Log–log plot of the transmitted information resource vs. the input size $n$ for solving Hidden Matching within error probability $p_{\text{error}} = 0.1$. We compare the best-known classical protocol, the classical lower bound, and the quantum coherent state protocol in the ideal setting, in the practical setting with the experimental parameters of Table 1, as well as in the post-selected case where Bob only outputs an outcome when he observes at least one click in the protocol run. The optimal mean photon number to obtain an error probability of 0.1 is $|\alpha|^2_{\text{ideal}} \approx 1.6$ whereas $|\alpha|^2_{\text{exp}} \approx 7.1$. The minimum input size needed for the coherent protocol to beat the classical protocol in the ideal, practical, and post-selected cases is $n = 17/2926/1760$, respectively. To beat the classical lower bound, the minimum input size for the coherent protocol in the ideal setting is $n = 10189$, whereas taking into account the experimental imperfections, $n = 394272$

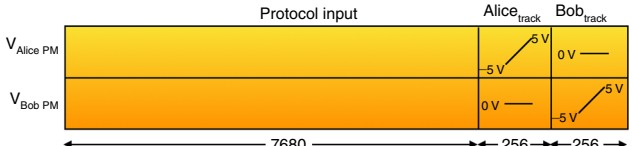

**Fig. 9** Block illustration for analyzing and correcting the phase drift in the pulse sequences of Alice and Bob. Phase tracking is done once for every block of 8192 pulses. The first 7680 pulses are used for performing the protocol. The second part of the block Alice$_{\text{track}}$, tracks the phase drift in Alice's PM. For this we give a ramp voltage from $-5$V to $+5$V to Alice's PM and 0 V to Bob's PM. The third part of the block Bob$_{\text{track}}$, tracks Bob's PM by giving a ramp voltage from $-5$V to $+5$V to Bob's PM and 0 V to Alice's PM

threshold input sizes as in Sampling Matching hold. Reaching these thresholds with the setup of Fig. 7 is currently beyond experimental reach. This motivates the definition of the Sampling Matching problem.

**Phase-correction procedure.** For our experiments for Sampling Matching, we have applied an averaging technique over blocks of pulses to correct for the phase drift occurring between Alice's and Bob's pulse sequences. Such an averaging corresponds well to our conditions, with the relatively high 1-MHz repetition rate of our experiment and the high stability of our setup. We make blocks of pulses and track the average of the phase drift in one block to use it to correct the drift of the subsequent block. The block construction we use is detailed in Fig. 9. We choose a block size of 8192 pulses. The first 7680 pulses are used for protocol run. The second segment of the block, Alice$_{\text{track}}$, tracks the phase drift in the path corresponding to Alice's PM. This is done by providing a ramp voltage in Alice's PM from $-5$V to $+5$V, and 0 V in Bob's PM across 256 pulses. The response of the linear ramp voltage across a phase modulator is a cosine function $A\cos(\omega t + \phi)$, which is tracked using the photodiode PD. We then model the expected response corresponding to the actual response, hence obtaining the information on the phase and the phase drift up to a certain error. If $V_{\text{bias}}$ is the voltage corresponding to the phase drift, then we add this factor to the voltage provided to Alice's PM for the next block, i.e., $V_{\text{PM}} = V_{x_i} + V_{\text{bias}}$. We similarly track and correct the phase drift in Bob's PM over the last 256 pulses of the block, Bob$_{\text{track}}$.

## Data availability

All relevant data are available from the authors upon request.

## Code availability

All relevant code is available from the authors upon request.

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

## Acknowledgements

We thank Frédéric Grosshans, Norbert Lütkenhaus, Luis Trigo Vidarte, and Adeline Orieux for useful discussions. This research was supported by the European Research Council projects QCC (I.K.) and QUSCO (E.D.), the French National Research Agency project quBIC and the BPI France project RISQ.

## Author contributions

All authors contributed to the theoretical analysis and the design of the experiment. N.K. developed the software, experimental system control tools, and collected the data. I.K. and E.D. conceived and supervised the project. All authors contributed to writing the paper.

## Additional information

**Competing interests:** The authors declare no competing interests.

