## [Peer Review File · Nature Communications]

Reviewers' Comments:

Reviewer #1:

Remarks to the Author:

This is an accessible and very well-written paper about experimentally demonstrating a quantum advantage in communication complexity. More specifically, a slight variant of the well-known Hidden Matching problem is introduced - termed Sampling Matching problem - and implemented with elements of quantum optics. The advantage of this novel variation being that much larger instances of the problem could be implemented, which then indeed allowed to beat the best known classical protocols in terms of the information transmitted. The results are novel and the paper is technically sound. The implementation was, however, not able to beat the (unconditional) classical lower bounds. Moreover, contrary to the theoretical results on the Hidden Matching problem, the quantum advantage in the optical implementation only concerns the transmitted information but not the use of the communication resource itself.

One can certainly argue that the question about quantum computational supremacy/advantage is a pressing one. However, the authors do not discuss a quantum advantage in computation in their submission but rather a quantum advantage in communication complexity - and in a very specific model at that. If we talk about quantum advantage in such a wide (non-computational) sense then, e.g., every violation of a Bell inequality or quantum key distribution scheme corresponds to an unbounded quantum advantage. Many more settings in quantum information come to mind, where a quantum advantage has been studied previously. In that sense, I fail to see how this submission is of specific, novel interest when it comes to demonstrating a quantum advantage. Hence, overall I am not sure how important the results are to the scientists working on quantum communication complexity or quantum optics. Whereas the paper clearly make some progress towards implementing quantum technologies, I am not convinced that results are of sufficient interest to warrant publication in Nature Communications.

Reviewer #2:

Remarks to the Author:

The manuscript entitled "Experimental demonstration of quantum advantage for one-way communication complexity" designed an experiment of Sampling Matching problem. The sampling matching problem is mathematically equivalent to the well known Hidden Matching problem, which can find immediate applications in quantum money and other quantum tasks.

The original Hidden Matching problem require Bob gives correct answer with a randomly given pair, which is hard to to be implemented. The author use the idea of passive Round-Robin DPS QKD, propose the concept of Sampling Matching problem, to show its equivalent to original Hidden Matching problem with classical method, and clearly demonstrate the quantum advantage. The paper is well written and the results are scientifically reasonable. Recently, quantum communication complexity has drawn more and more research interest. In that sense, this paper is on time. Therefore, I recommend its publication on Nature Communication after the authors adress the below comments.

1. This work is still a proof-of-principle experiment because Alice and Bob shares same laser, and the distance between Alice and Bob is zero. They may both limit its future application. The author should clarify it in the abstract or in the beginning of the paper.

2. The cited references is kind of limited. For example, the authors cite their own coin flipping experiment but do not cite the bit commitment experiment by other groups, like PRL 111, 180504

(2013); PRL 112, 010504 (2014). Also, the recent experimental device independent quantum random number generation, which is an good example of quantum victory based on Bell inequality violation, has not been cited.

Reviewer #4:

Remarks to the Author:

The main proposal of the paper is based on a quite well-known scheme for unambiguous discrimination of the parity of two binary phase shift keying (BPSK) modulated coherent state using a 50-50 beamsplitter:

1. $|-a\rangle$ and $|-a\rangle$ fed into a beamsplitter gives $|\sqrt{2}a\rangle$ and $|0\rangle$
2. $|+a\rangle$ and $|+a\rangle$ fed into a beamsplitter gives $|\sqrt{2}a\rangle$ and $|0\rangle$
3. $|-a\rangle$ and $|+a\rangle$ fed into a beamsplitter gives $|0\rangle$ and $|\sqrt{2}a\rangle$
4. $|+a\rangle$ and $|-a\rangle$ fed into a beamsplitter gives $|0\rangle$ and $|\sqrt{2}a\rangle$

If options 1 or 2 happen (both inputs have same sign), detector 1 clicks. If options 3 or 4 happen (the two inputs have opposite sign), detector 2 clicks.

The authors use the observation above to propose a protocol where Alice provides an n bit string in the form of n BPSK coherent states: $|+a\rangle|-a\rangle\dots|-a\rangle$, and the Bob picks the k -th state and the l -th state, mixes them on a beamsplitter and detects both outputs, and by doing so, modulo a probability of erasure (probability the $|\sqrt{2}a\rangle$ or $|\sqrt{2}a\rangle$ does not generate a click, which happens with probability $\exp(-2|a|^2)$), gets the parity of the k -th and l -th bit pair.

The connection of the above proposal to the Hidden Matching problem that is the main subject matter of the paper, and all the complexity arguments as to why this proposed protocol is better than the classical strategies, and the relationship to the qubit protocol, are unclear to me.

The paper's main result may be interesting, but I do not recommend publication in its current form, since I cannot understand the main result (beyond the above well-known fact about USD measurement of BPSK coherent state parity). I am willing to review another revision of the paper, if the paper is re-written to explain things better. Please see some comments below.

Detailed comments:

Introduction

"require a large number of active components increasing with the input size of the problem." -- what is large? The previous sentence uses the phrase "constant", the performance of the quantum algorithm should be quantified properly.

Results

The Hidden Matching problem, on which the entire paper's results are based, is not properly defined. Matching is usually defined for a graph G as a set of edges of G such that no two share a vertex in common. A graph with n nodes has a perfect matching when there are $n/2$ edges in the matching. Not all graphs has perfect matchings. What do the authors mean by "matching" in this paper? Are these matchings of a clique? I suggest defining the term matching in the context of the paper, and then define what is meant by "edge disjoint", and thereby discuss why there should be $n-1$ unique edge-

disjoint matchings in the stated problem. The authors are having the reader infer what these things mean from a diagram in Figure 2, and it is not at all clear -- at least to me -- either from the diagram or the text what the problem is.

What is the objective of the problem? I take that Bob receives ONE matching -- a set of edges of a fully-connected graph of n nodes with no two edge sharing a common vertex -- and that he has to output ALL the parities $x_k + x_l$ for all edges (k, l) that are in the matching given to him?

"communication and information resources" is ill defined. Communication resource, Computation resource and Storage resource are well-understood terms, and there is a lot of literature on tradeoffs between these for different multiparty computing protocols. I don't know what is meant by "information resource".

I skipped the classical strategy section, as I didn't follow the problem statement, I couldn't follow the scaling arguments for the classical algorithms.

The qubit protocol is clear, but the connection to the hidden matching problem is not entirely clear to me. Here, $|x\rangle$ is encoded into $\log n$ qubits, and by one (destructive) measurement, Bob gets a *single* parity, that of the k -th and the l -th bits. However, I don't understand the sentence, "both the communication and the transmitted information are exponentially better than in the classical case." How is "communication" different from "transmitted information"?

Coherent state protocol: Here the authors leverage a protocol proposed by Arrazola and Lutkenhaus for checking equality of single bits held by two parties. The annihilation operator a_x does not have the right commutator - a normalization is needed since x_k 's are just 0s or 1s.

Reply to Reviewers

We are grateful to all Reviewers for their comments and suggestions that helped to substantially improve our manuscript.

Reviewer #1:

This is an accessible and very well-written paper about experimentally demonstrating a quantum advantage in communication complexity. More specifically, a slight variant of the well-known Hidden Matching problem is introduced - termed Sampling Matching problem - and implemented with elements of quantum optics. The advantage of this novel variation being that much larger instances of the problem could be implemented, which then indeed allowed to beat the best known classical protocols in terms of the information transmitted. The results are novel and the paper is technically sound. The implementation was, however, not able to beat the (unconditional) classical lower bounds. Moreover, contrary to the theoretical results on the Hidden Matching problem, the quantum advantage in the optical implementation only concerns the transmitted information but not the use of the communication resource itself.

We thank the Reviewer for their comments on the accessibility and soundness of the paper. The Reviewer rightly mentions that our optical implementation allows for an advantage in transmitted information and not in the communication resource itself. However, this is an inherent property of all optical implementations using coherent states and it is not a drawback specifically of our implementation or of the problem we implement. Even though an advantage both in communication and in transmitted information would be ideal, this remains out of reach for current state-of-the-art technologies (and it is very probable it will remain out of reach for years to come). Hence, our implementation offers the best we can hope with the state-of-the-art technology, and that for the first time and only after having to introduce a novel theoretical problem precisely to make the implementation possible.

One can certainly argue that the question about quantum computational supremacy/advantage is a pressing one. However, the authors do not discuss a quantum advantage in computation in their submission but rather a quantum advantage in communication complexity - and in a very specific model at that. If we talk about quantum advantage in such a wide (non-computational) sense then, e.g., every violation of a Bell inequality or quantum key distribution scheme corresponds to an unbounded quantum advantage. Many more settings in quantum information come to mind, where a quantum advantage has been studied previously. In that sense, I fail to see how this submission is of specific, novel interest when it comes to demonstrating a quantum advantage. Hence, overall I am not sure how important the results are to the scientists working on quantum communication complexity or quantum optics. Whereas the paper clearly make some progress towards implementing quantum technologies, I am not convinced that results are of sufficient interest to warrant publication in Nature Communications.

It is true that our quantum advantage pertains to the communication complexity model and not to computation. It is also true that advantages in different models have been demonstrated before, for example in non-locality and in QKD experiments. We still believe however that experimentally demonstrating an advantage in a new, important model, where it has not been demonstrated ever before, pushes the boundaries of what is possible in quantum technologies and brings us closer to useful applications. Non-locality experiments remained very interesting even though QKD experiments had been demonstrated before, and similarly we believe the first ever demonstration of an advantage in one-way communication complexity remains very interesting even though non-locality and QKD experiments have been performed before.

Let us also comment on the importance of these results to the “scientists working on quantum communication complexity”. One can cite three main results in quantum communication complexity, one for each of the main models: Quantum fingerprints [H. Buhrman, R. Cleve, J. Watrous and R. de Wolf, Phys. Rev. Lett. 87, 167902 (2001)], which provided the first exponential advantage in the model of simultaneous messages; the Hidden Matching papers, from one of the authors of the current paper, which appeared in the most prestigious conference in Theoretical Computer Science (STOC) in two different years as well as in the most important conference on quantum information (QIP); and a work by Ran Raz, which provides a separation in the two-way communication model, again presented in STOC and QIP. All these works have received a very big number of citations. We can see that these separations are hard to find and are most highly regarded in the wider community. Finding ways to implement these protocols and demonstrate these advantages has been open for more than a decade. Implementations in the Simultaneous Message model appeared four years ago using linear optics and the results were published in Nature Communications [F. Xu et al., Nature Communications 6, 8735 (2015)] (and a second implementation in PRL [J. Guan et al. Phys. Rev. Lett. 116, 240502]). Since then, demonstrating such an advantage in the stronger model of one-way communication was open and seemed out of reach. We managed to provide such an advantage by novel theoretical work in defining a new problem and providing rigorously bounds on its complexity, and by using state-of-the-art linear optics to provide an implementation. We strongly believe our results have considerably moved forward the state-of-the-art and this has been acknowledged already by the wider community and the media.

Our results on quantum advantage in one-way communication complexity also have implications in verification and cryptographic tasks. The widely studied Hidden Matching problem has been used to propose quantum money protocols to demonstrate unconditional security [IEEE, PRL, PRA]. This has generated tremendous interest recently. However the experimental demonstration of Hidden Matching problem has been out of reach of the current technologies primarily because the implementation circuit becomes more complex with increasing input sizes. Our optical implementation of the Sampling Matching problem requires a fixed circuit, irrespective of the input size, and hence is much easier to implement experimentally. Similar to the original Hidden Matching problem, our problem can be used for the different verification and cryptographic tasks and hence this paves the way towards their experimental implementation in the future attesting to the potential impact of our work for future directions of research showing a quantum advantage in different models.

Reviewer #2:

The manuscript entitled “Experimental demonstration of quantum advantage for one-way communication complexity” designed an experiment of Sampling Matching problem. The sampling matching problem is mathematically equivalent to the well known Hidden Matching problem, which can find immediate applications in quantum money and other quantum tasks.

The original Hidden Matching problem require Bob gives correct answer with a randomly given pair, which is hard to to be implemented. The author use the idea of passive Round-Robin DPS QKD, propose the concept of Sampling Matching problem, to show its equivalent to original Hidden Matching problem with classical method, and clearly demonstrate the quantum advantage. The paper is well written and the results are scientifically reasonable. Recently, quantum communication complexity has drawn more and more research interest. In that sense, this paper is on time. Therefore, I recommend its publication on Nature Communication after the authors address the below comments.

1. This work is still a proof-of-principle experiment because Alice and Bob shares same laser, and the distance between Alice and Bob is zero. They may both limit its future application. The author should clarify it in the abstract or in the beginning of the paper.

2. The cited references is kind of limited. For example, the authors cite their own coin flipping experiment but do not cite the bit commitment experiment by other groups, like PRL 111, 180504 (2013); PRL 112, 010504 (2014). Also, the recent experimental device independent quantum random number generation, which is an good example of quantum victory based on Bell inequality violation, has not been cited.

We thank the Reviewer for their positive recommendation and their helpful comments. We have addressed the points the Reviewer raises in our revised manuscript. In particular, we have added 'proof-of-principle' in the abstract and we have added more references for experimental demonstrations of quantum cryptographic primitives and quantum random number generation (Refs [21], [22] and [27]).

Reviewer #4:

The main proposal of the paper is based on a quite well-known scheme for unambiguous discrimination of the parity of two binary phase shift keying (BPSK) modulated coherent state using a 50-50 beamsplitter:

1. $| -a \rangle$ and $| -a \rangle$ fed into a beamsplitter gives $| -\sqrt{2}a \rangle$ and $| 0 \rangle$
2. $| +a \rangle$ and $| +a \rangle$ fed into a beamsplitter gives $| \sqrt{2}a \rangle$ and $| 0 \rangle$
3. $| -a \rangle$ and $| +a \rangle$ fed into a beamsplitter gives $| 0 \rangle$ and $| -\sqrt{2}a \rangle$
4. $| +a \rangle$ and $| -a \rangle$ fed into a beamsplitter gives $| 0 \rangle$ and $| \sqrt{2}a \rangle$

If options 1 or 2 happen (both inputs have same sign), detector 1 clicks. If options 3 or 4 happen (the two inputs have opposite sign), detector 2 clicks.

The authors use the observation above to propose a protocol where Alice provides an n bit string in the form of n BPSK coherent states: $| +a \rangle | -a \rangle \dots | -a \rangle$, and the Bob picks the k -th state and the l -th state, mixes them on a beam-splitter and detects both outputs, and by doing so, modulo a probability of erasure (probability the $| -\sqrt{2}a \rangle$ or $| +\sqrt{2}a \rangle$ does not generate a click, which happens with probability $\exp(-2|a|^2)$), gets the parity of the k -th and l -th bit pair.

The connection of the above proposal to the Hidden Matching problem that is the main subject matter of the paper, and all the complexity arguments as to why this proposed protocol is better than the classical strategies, and the relationship to the qubit protocol, are unclear to me.

The paper's main result may be interesting, but I do not recommend publication in its current form, since I cannot understand the main result (beyond the above well-known fact about USD measurement of BPSK coherent state parity). I am willing to review another revision of the paper, if the paper is re-written to explain things better. Please see some comments below.

We thank the Reviewer for their critical assessment of our paper. As we explain below in our answers to the detailed comments, we have substantially revised especially the first part of our manuscript to provide more explanations and clarifications on the protocol that we have implemented.

Detailed comments:

Introduction

"require a large number of active components increasing with the input size of the problem." -- what is large? The previous sentence uses the phrase "constant", the performance of the quantum algorithm should be quantified properly.

The number of active components in the Hidden Matching implementation using the coherent state mapping grows at least logarithmically with the input size, which would make an implementation completely out of reach. For example, for beating the best classical protocol we needed the size n to be around 4000. The logarithm of that is more than 60, hence one would need to align more than 60 active delay lines to the same error as our scheme of one component (one beam splitter) in order to beat the best classical protocol, which is completely impractical. Our new scheme, as can be seen in particular in Fig. 6, uses a single beam splitter before measuring the pulses.

We have now clarified this point in the revised manuscript.

Results

The Hidden Matching problem, on which the entire paper's results are based, is not properly defined. Matching is usually defined for a graph G as a set of edges of G such that no two share a vertex in common. A graph with n nodes has a perfect matching when there are $n/2$ edges in the matching. Not all graphs has perfect matchings. What do the authors mean by "matching" in this paper? Are these matchings of a clique? I suggest defining the term matching in the context of the paper, and then define what is meant by "edge disjoint", and thereby discuss why there should be $n-1$ unique edge-disjoint matchings in the stated problem. The authors are having the reader infer what these things mean from a diagram in Figure 2, and it is not at all clear -- at least to me -- either from the diagram or the text what the problem is.

We thank the Reviewer for this important comment. We have added a significantly more detailed description of the problem which we believe clarifies the setting.

"The Hidden Matching problem is illustrated in Fig.1. It is a one-way communication complexity task involving two players, Alice and Bob, and is described as follows. For any positive even integer n , Alice receives as input a string $x \in \{0,1\}^n$ while Bob receives a perfect matching σ_i on the complete graph (where all edges are present) of n vertices (where the vertices are indexed with the numbers $\{1,2,\dots,n\}$) uniformly at random from a set of $n-1$ edge-disjoint perfect matchings $\mathcal{M}_n \in \{\sigma_1, \dots, \sigma_{n-1}\}$. A perfect matching here is a list of $n/2$ pairs of vertices such that no vertex appears twice in the list. A set of edge-disjoint perfect matchings is a set of matchings where each edge (pair of vertices) appears at most once in the set. It is well known that a clique of n nodes can be decomposed into a set of $n-1$ edge-disjoint perfect matchings. This set is known to both Alice and Bob, and Bob receives as input one of the matchings in the set."

What is the objective of the problem? I take that Bob receives ONE matching -- a set of edges of a fully-connected graph of n nodes with no two edge sharing a common vertex -- and that he has to output ALL the parities $x_k + x_l$ for all edges (k, l) that are in the matching given to him?

Again we thank the Reviewer for asking for the clarification of this crucial point.

In the Hidden Matching problem, Bob indeed receives one perfect matching uniformly at random from the set of matchings. The matching Bob receives is unknown to Alice. Using our example of $n=4$ and the three matchings M_1, M_2, M_3 (as in Fig. 2), let's say Bob receives the matching M_1 . Then the objective of the problem is the following: from the message Alice sends to Bob, Bob has to correctly

output the parity of any one of the pairs of matching M_1 . So he can choose to output either pair (1,2) and the parity $x_1 + x_2$, or he can choose to output the pair (3,4) and output the parity $x_3 + x_4$. We have clarified this point in the revised manuscript.

"communication and information resources" is ill defined. Communication resource, Computation resource and Storage resource are well-understood terms, and there is a lot of literature on tradeoffs between these for different multiparty computing protocols. I don't know what is meant by "information resource".

We have added explicit definitions to the revised manuscript as follows.

"In this [one-way communication] model, the communication cost of a protocol is the number of bits Alice has to send to Bob in order to solve the problem, while the transmitted information, instead of the number of bits sent, calculates the real bits of information about the inputs that the messages carry. For example, if Alice always sends the same, long message, independent of her input, then the communication cost will be large, while the transmitted information will be zero, since no information about her input has been transmitted. Transmitted information is a resource that is important for privacy, when on top of having an efficient protocol, we want the players to solve the task without learning much about the other player's input. One can define the transmitted information as the mutual information between the messages and the inputs and can upper bound it with the logarithm of the number of different messages. The transmitted information is always at most the communication cost, since one bit carries at most one bit of information, and hence the bottleneck is always the time."

I skipped the classical strategy section, as I didn't follow the problem statement, I couldn't follow the scaling arguments for the classical algorithms.

We have added more details in this section of the manuscript as well. Note that this part does not contain new results so we refrained from giving all details. We provide the high level ideas and refer the reader to the corresponding published papers.

*The qubit protocol is clear, but the connection to the hidden matching problem is not entirely clear to me. Here, $|x\rangle$ is encoded into $\log n$ qubits, and by one (destructive) measurement, Bob gets a *single* parity, that of the k -th and the l -th bits. However, I don't understand the sentence, "both the communication and the transmitted information are exponentially better than in the classical case." How is "communication" different from "transmitted information"?*

In the quantum protocol, Bob receives the state $|x\rangle$ and performs a measurement that gives him a single parity, which is what needed to solve the Hidden Matching problem. The protocol only communicates $\log n$ qubits so both the communication and the transmitted information are at most $\log n$ (the transmitted information is always at most the communication; we have now defined both notions before in the manuscript as explained above). On the other hand, what has been proven in Ref. [34] is that the optimal classical protocol for this problem must communicate \sqrt{n} bits and also must have transmitted information at least \sqrt{n} . Thus we have an exponential difference in both resources ($\log n$ vs. \sqrt{n}).

Note that, the transmitted information and the communication of a protocol need not be always the same. For example a protocol where Alice sends 100 bits always equal to 0, has communication 100, though the transmitted information is 0, since Bob did not learn anything he didn't already know, since the message is always fixed to be the 0 string. This is why these two resources cannot be treated interchangeably.

Coherent state protocol: Here the authors leverage a protocol proposed by Arrazola and Lutkenhaus for checking equality of single bits held by two parties. The annihilation operator a_x does not have the right commutator - a normalization is needed since x_k 's are just 0s or 1s.

We thank the Reviewer for pointing out this normalization missing factor in the coherent state protocol. We have now corrected it.

We sincerely hope that the clarifications of our setting and protocol that we have provided in the revised manuscript also highlight the impact of our work. By proposing a new problem with the same complexity as the original one but tailored to an experimental implementation, we have managed to use state-of-the-art quantum optics technology based on notions involving coherent state parity measurement as the Reviewer has remarked, to demonstrate quantum advantage for a task that was previously considered inaccessible.

Reviewers' Comments:

Reviewer #1:

Remarks to the Author:

For this second round of refereeing the authors partially rewrote their manuscript and carefully replied to all the comments made. In response to my report the authors mainly argue the following points:

- 1) quantum communication complexity is an important and difficult research area, with additional applications in verification and cryptography
- 2) implementing theoretical work on classical-quantum complexity gaps has so far mostly been elusive
- 3) the work under review is what is possible with current state-of-the-art technology

Whereas I do not have a good overview about works on implementations, I do not think that I question any of these points. I also acknowledge that the authors have some good arguments. Nevertheless, as indicated in my previous report, it is not fully clear to me that the theory part of the current work is strong enough for publication in Nature Communications. However, if the other referees confirm that the submission represents strong work on the experimental side, then I am overall okay with acceptance.

Reviewer #5:

Remarks to the Author:

In this paper the authors propose and experimentally demonstrate a new communications task: the sampling matching problem, an interesting twist on the hidden matching problem. Similar to the hidden matching problem, also the sampling matching features an exponential gap in the one-way communication model. However, the authors show that the sampling matching problem admits a simple implementation with quantum optical elements as opposed to the hidden matching problem which seems impractical for experimental implementation. The authors go beyond this already strong theory results and provide a proof of principle implementation beating the best known classical protocol.

In my opinion, this is one of the most interesting works of the year. It combines novel relevant theory --the sampling matching problem will find broad application in CS and crypto-- with a proof of principle experimental demonstration. My only concern, and it aligns with comments in the previous review round, relates to the presentation and the claims of quantum advantage. While there is a clear theoretical quantum advantage: the sampling matching problem features an exponential gap, the experimental implementation is only able to beat the best known classical protocol. In my opinion, the community has already suffered enough from overclaims. I would suggest acceptance provided that the claims in the title and abstract are modified to acknowledge that the advantage is not absolute and only refers to the best known protocol.

Reply to Reviewers

We are grateful to all Reviewers for their helpful comments and suggestions.

Reviewer #1:

For this second round of refereeing the authors partially rewrote their manuscript and carefully replied to all the comments made. In response to my report the authors mainly argue the following points:

- 1) quantum communication complexity is an important and difficult research area, with additional applications in verification and cryptography*
- 2) implementing theoretical work on classical-quantum complexity gaps has so far mostly been elusive*
- 3) the work under review is what is possible with current state-of-the-art technology*

Whereas I do not have a good overview about works on implementations, I do not think that I question any of these points. I also acknowledge that the authors have some good arguments. Nevertheless, as indicated in my previous report, it is not fully clear to me that the theory part of the current work is strong enough for publication in Nature Communications. However, if the other referees confirm that the submission represents strong work on the experimental side, then I am overall okay with acceptance.

We thank the Reviewer for their positive recommendation.

Reviewer #5:

In this paper the authors propose and experimentally demonstrate a new communications task: the sampling matching problem, an interesting twist on the hidden matching problem. Similar to the hidden matching problem, also the sampling matching features an exponential gap in the one-way communication model. However, the authors show that the sampling matching problem admits a simple implementation with quantum optical elements as opposed to the hidden matching problem which seems impractical for experimental implementation. The authors go beyond this already strong theory results and provide a proof of principle implementation beating the best known classical protocol.

In my opinion, this is one of the most interesting works of the year. It combines novel relevant theory - the sampling matching problem will find broad application in CS and crypto-- with a proof of principle experimental demonstration. My only concern, and it aligns with comments in the previous review round, relates to the presentation and the claims of quantum advantage. While there is a clear theoretical quantum advantage: the sampling matching problem features an exponential gap, the experimental implementation is only able to beat the best known classical protocol. In my opinion, the community has already suffered enough from overclaims. I would suggest acceptance provided that the claims in the title and abstract are modified to acknowledge that the advantage is not absolute and only refers to the best known protocol.

We thank the Reviewer for acknowledging the novelty and potential impact of our work and for their positive recommendation. We do sincerely agree that overclaiming is a bad habit for the field and we believe we have made clear that we are demonstrating advantage with respect to the best classical protocol in several places in the manuscript, in particular in the abstract and introduction but not in the title due to space constraints.